# Effect of Chitosan Degradation Products, Glucosamine and Chitosan Oligosaccharide, on Osteoclastic Differentiation

**DOI:** 10.3390/biotech13010006

**Published:** 2024-03-06

**Authors:** Tomoharu Takeuchi, Midori Oyama, Tomomi Hatanaka

**Affiliations:** 1Faculty of Pharmacy and Pharmaceutical Sciences, Josai University, 1-1 Keyakidai, Sakado-shi 350-0295, Japan; oyamami@josai.ac.jp (M.O.); tmmhtnk@josai.ac.jp (T.H.); 2School of Medicine, Tokai University, 143 Shimokasuya, Isehara 259-1193, Japan

**Keywords:** osteoclast, glucosamine, chitosan oligosaccharide, chitosan, bone, biomaterial

## Abstract

Chitosan, a natural cationic polysaccharide derived from crustaceans and shellfish shells, is known for its advantageous biological properties, including biodegradability, biocompatibility, and antibacterial activity. Chitosan and its composite materials are studied for their potential for bone tissue repair. However, the effects of chitosan degradation products, glucosamine (GlcN) and chitosan oligosaccharide (COS), on osteoclasts remain unclear. If these chitosan degradation products promote osteoclastic differentiation, careful consideration is required for the use of chitosan and related materials in bone repair applications. Here, we assessed the effects of high (500 μg/mL) and low (0.5 μg/mL) concentrations of GlcN and COS on osteoclastic differentiation in human peripheral blood mononuclear cells (PBMCs) and murine macrophage-like RAW264 cells. A tartrate-resistant acid phosphatase (TRAP) enzyme activity assay, TRAP staining, and actin staining were used to assess osteoclastic differentiation. High concentrations of GlcN and COS, but not low concentrations, suppressed macrophage colony-stimulating factor (M-CSF)- and RANKL-dependent increases in TRAP enzyme activity, TRAP-positive multinuclear osteoclast formation, and actin ring formation in PBMCs without cytotoxicity. Similar effects were observed in the RANKL-dependent osteoclastic differentiation of RAW264 cells. In conclusion, chitosan degradation products do not possess osteoclast-inducing properties, suggesting that chitosan and its composite materials can be safely used for bone tissue repair.

## 1. Introduction

Chitosan is a naturally occurring cationic polysaccharide obtained by the deacetylation of chitin, which is found mainly in the exoskeletons of crustaceans and insects [1]. Chitin is a linear polysaccharide consisting of β-(1-4)-linked *N*-acetyl-_D_-glucosamine (GlcNAc) units, while chitosan consists mainly of _D_-glucosamine (GlcN) units [1]. Chitosan is widely recognized for its advantageous properties such as biocompatibility, low antigenicity, and biodegradability. Chitosan’s versatility finds expression in diverse fields, with applications ranging from drug delivery systems, wound healing, and tissue engineering [2,3]. In tissue engineering, chitosan scaffolds provide a supportive environment for cell adhesion, proliferation, and differentiation, thereby facilitating the creation of functional tissues. Their antimicrobial properties further contribute to the maintenance of a sterile environment during tissue regeneration processes [2,3,4]. With these attributes, chitosan and its composite materials hold significant promise for advancing bone tissue engineering and contributing to the development of innovative solutions for regenerative medicine and biomedical research [5,6,7,8].

Bone, a structurally and functionally critical component of the body, necessitates advancements in treatment methods for fractures and injuries [9]. Understanding the intricate cellular processes of bone metabolism involving osteoclasts and osteoblasts is of paramount importance in the field of bone tissue engineering [5,6,7,8]. Osteoclasts are responsible for bone resorption, whereas osteoblasts contribute to new bone formation. Effectively controlling and inducing these processes are key factors for the successful engineering of bone tissue [8]. Osteoclasts are multinucleated cells that play a pivotal role in bone resorption and are differentiated from monocyte/macrophage lineage cells upon the stimulation with macrophage colony-stimulating factor (M-CSF) and receptor activator of nuclear factor-κB (NF-κB) ligand (RANKL) [10,11]. M-CSF promotes the expression of receptor activator of nuclear factor-κB (RANK) which is a receptor of RANKL. Upon binding RANKL to its receptor RANK, a series of intracellular events are initiated, including the activation of mitogen-activated protein kinases (MAPKs), such as p38, ERK, and JNK, which play crucial roles in the regulation of osteoclast differentiation [10]. Simultaneously, the RANKL-RANK interaction induces the activation of the NF-κB signaling pathway. NF-κB activation leads to the translocation of NF-κB complexes into the nucleus, where they regulate the transcription of genes important for osteoclasts. Following the activation of these early signaling pathways, sequential events occur, leading to the differentiation of the cells into osteoclasts: the expression of osteoclast marker genes, including tartrate-resistant acid phosphatase (TRAP); cell fusion and the formation of TRAP-positive multinuclear cells; actin ring formation, serving as a structure for osteoclasts to adhere tightly to the surrounding bone surface; and bone resorption [10].

When chitosan is used in bone tissue engineering, it undergoes biodegradation mainly by lysozyme within the tissue, breaking down into components such as chitosan oligosaccharide (COS) with up to 20 mer of GlcN unit [12]. Furthermore, the fundamental unit of chitosan, glucosamine (GlcN), is among the degradation products. However, the specific effects of these degradation products on osteoclasts remain unclear. Chitosan has been reported to suppress osteoclast differentiation [13]. A high concentration (440 μg/mL) of COS has also been reported to suppress TRAP-positive multinuclear cell formation and bone resorption activity in in vitro mouse primary osteoclast cultures [14]. Meanwhile, low concentrations of COS (up to 0.5 μg/mL) have been reported to promote the expression of osteoclast marker genes and the formation of TRAP-positive multinuclear cells in vitro [15]. GlcN has also been reported to suppress osteoclast differentiation in vitro using murine RAW264 cells [16], although its effect on osteoclasts at low concentrations such as 0.5 μg/mL is unreported so far. If the chitosan degradation products GlcN and COS promote osteoclastic differentiation, careful consideration is required for the use of chitosan and related materials in bone repair applications since the degradation of chitosan may potentially delay bone regeneration. However, the effects of those chitosan degradation products on osteoclasts remain unclear: the impact of these aminosaccharides on osteoclast differentiation may be concentration-dependent or cell-type-specific.

In this study, in order to clarify the effect of chitosan degradation products on osteoclast differentiation, we assessed and compared the effects of high (500 μg/mL) and low (0.5 μg/mL) concentrations of GlcN and COS on osteoclastic differentiation. These concentrations were chosen based on the previously published manuscripts: since the treatment with a low concentration of COS (0.5 μg (500 ng)/mL) was reported to promote osteoclast differentiation [15], we chose 0.5 μg/mL as the low concentration; on the other hand, a high concentration (440 μg/mL) of COS was reported to suppress osteoclast differentiation [14,15]. As a neat and rounded concentration, we chose 500 μg/mL as the high concentration. Additionally, we assessed the effect of those aminosaccharides in two model systems because the effect of the saccharides might be different between the cells used: the primary culture consisted of human peripheral blood mononuclear cells (PBMCs) which were used as a precursor of human osteoclasts, and the model cell line murine RAW264 cells which are widely used simple model cells for osteoclast differentiation [17,18]. 

## 2. Materials and Methods

### 2.1. Cell Culture

Non-characterized normal human PBMCs were obtained from Fujifilm Wako (Osaka, Japan) and cultured in minimum essential medium alpha (MEMα) (Fujifilm Wako) containing 10% heat-inactivated fetal bovine serum (Thermo Fisher Scientific, Waltham, MA, USA) and 1× penicillin/streptomycin (Fujifilm Wako). The mouse macrophage-like RAW264 [19] cell line was obtained from the RIKEN Cell Bank (Tsukuba, Japan) and maintained in MEMα medium containing 10% heat-inactivated fetal bovine serum and 1× penicillin/streptomycin under a humidified atmosphere containing 5% CO_2_ at 37 °C. 

### 2.2. Osteoclastic Differentiation

Human PBMCs were seeded 1 × 10^5^ live cell/well on a 96-well plate (Thermo Fisher Scientific) or a 96-well black plate (Corning, Corning, NY, USA). Then, the cells were treated with 100 ng/mL soluble RANKL (sRANKL) (Oriental Yeast, Tokyo, Japan) and 25 ng/mL human M-CSF (PeproTech, Rocky Hill, NJ, USA) in the presence of 0.5 or 500 μg/mL of _D_(+)-glucosamine hydrochloride (GlcN; Fujifilm Wako) or chitosan oligosaccharide (COS, degree of polymerization (DP) = 2–6; Tokyo Chemical Industry (TCI), Tokyo, Japan) and allowed to differentiate for 10 days, during which monocytes were differentiated to macrophages and subsequently to osteoclasts. The medium was replenished every three or four days. The saccharides were dissolved in phosphate-buffered saline (PBS; 8.1 mM Na_2_HPO_4_, 1.47 mM KH_2_PO_4_, 137 mM NaCl, and 2.68 mM KCl; pH 7.4). The cells without sRANKL and M-CSF treatment were used as the negative control. The concentrations of the saccharide were chosen based on a previous report about COS [14,15]. 

Mouse macrophage RAW264 cells were seeded 1 × 10^3^ live cell/well in a 96-well plate or a 96-well black plate and cultured for 1 d. The cells were then treated with 250 ng/mL sRANKL in the presence of 0.5 or 500 μg/mL of GlcN or COS and allowed to differentiate for 4 days. The cells without sRANKL treatment were used as the negative control.

### 2.3. Cell Viability Assay

PrestoBlue^TM^ reagent (Thermo Fisher Scientific) or Cell Counting Kit-8 reagent (Dojindo, Kumamoto, Japan) was added to the medium of differentiated cells and incubated under a humidified atmosphere containing 5% CO_2_ at 37 °C for 1 h. Fluorescence (excitation 544 nm, emission 590 nm; PrestoBlue^TM^ reagent) or absorbance (450 nm; Cell Counting Kit-8 reagent) was measured using a SpectraMax M5 microplate reader (Molecular Devices, Sunnyvale, CA, USA). The analysis settings were as follows: automix, off; calibrate, on; C. speed, normal. The fluorescence or absorbance of the medium without cells was subtracted as a background, and the data were presented as relative values normalized to that of PBS treated with RANKL (and M-CSF).

### 2.4. TRAP Enzyme Activity Assay

Differentiated cells were washed with PBS and lysed with 100 μL TRAP buffer (50 mM sodium tartrate, 50 mM sodium acetate, 150 mM KCl, 0.1% TritonX-100, 1 mM sodium ascorbate, and 0.1 mM FeCl_3_, pH 5.2) for 10 min on ice. Next, 10 μL of the thus prepared cell extract was added to 100 μL TRAP buffer containing 2.5 mM *p*-nitrophenyl phosphate (Thermo Fisher Scientific) as a TRAP substrate, and the reaction mixture was incubated for 1 h at 37 °C. Then, 50 μL of 0.9 M NaOH was added to the mixture to stop the reaction, and the absorbance at 405 nm was measured using a SpectraMax M5 microplate reader (Molecular Devices) as described previously [20]. The absorbance of the sample without cells was subtracted as a background.

### 2.5. TRAP Staining

Differentiated cells were washed with PBS and treated with 4% paraformaldehyde solution (Fujifilm Wako) for 10 min at room temperature, approximately 24 ± 2 °C. The cells were washed again with PBS and then stained with a TRAP staining solution containing 50 mM sodium tartrate, 45 mM sodium acetate, pH 5.2, 0.1 mg/mL naphthol AS-MX phosphate (Sigma-Aldrich, St. Louis, MO, USA), and 0.6 mg/mL fast red violet LB (Sigma-Aldrich), pH 5.2, for 1 h or longer at room temperature. The stained cells were observed under an EVOS XL Core microscope (Thermo Fisher Scientific), and TRAP-positive cells that stained red and contained three or more nuclei were counted to avoid counting the cells ongoing division which could have two nuclei as described previously [20]. Images were captured using a 10× objective.

### 2.6. Actin Staining

Differentiated cells on a 96-well black plate were washed with PBS and treated with 4% paraformaldehyde solution for 10 min at room temperature, approximately 24 ± 2 °C. The cells were washed with PBS and treated with PBS containing 0.1% TritonX-100 for 5 min at room temperature. After washing with PBS, the cells were stained using ActinGreen^TM^ 488 ReadyProbes™ Reagent (Thermo Fisher Scientific) dissolved in PBS for 30 min at room temperature according to the manufacturer’s instructions. The cells were washed again with PBS and then stained with 1 μg/mL of -Cellstain^®^- DAPI solution (Dojindo) dissolved in PBS for 5 min at room temperature. Green and blue fluorescence emitted by the stained cells was observed and visualized using a BZ-X800 fluorescence microscope (Keyence, Osaka, Japan). Images were captured using a 4× objective lens. The area surrounded by an actin ring with three or more nuclei in each image was manually selected and analyzed using Image J software Ver 1.54d [21].

### 2.7. Statistical Analysis

Data are presented as the means ± S.D. The statistical analysis was performed using StatMate 3 software Ver 3.19 (ATOMS Inc., Tokyo, Japan). Assuming a normal distribution, the significance of variance was determined using one-way ANOVA, followed by Tukey’s multiple comparison test for comparing multiple groups. *p*-values < 0.05 were considered statistically significant. We conducted the experiments at least twice and consistently obtained statistically similar results.

## 3. Results

### 3.1. Effects of GlcN and COS on Cell Viability of Human PBMCs and Murine RAW264 Cells

We first investigated the effects of low (0.5 μg/mL) and high (500 μg/mL) concentrations of GlcN and COS on the viability of human PBMCs and murine RAW264 cells (Figure 1). Under the differentiation conditions used in this study, low and high concentrations of the aminosaccharides had no significant effects on cell viability.

### 3.2. Effects of GlcN and COS on TRAP Enzyme Activity of Human PBMCs and Murine RAW264 Cells

Next, we assessed the effect of saccharides on TRAP enzyme activity. The activity of human PBMCs increased after treatment with M-CSF and RANKL (Figure 2a). This increase was significantly reduced by the addition of high concentrations of GlcN or COS and was not affected by low concentrations of them. In RAW264 cells, we found that the RANKL-dependent increase in absorbance was significantly suppressed by the addition of high concentrations of GlcN, and high concentrations of COS tended to suppress TRAP enzyme activity (Figure 2). 

### 3.3. Effects of GlcN and COS on TRAP-Positive Multinuclear Cell Formation of Human PBMCs and Murine RAW264 Cells

The RANKL (and M-CSF)-dependent formation of TRAP-positive multinuclear cells was evaluated by TRAP staining (Figure 3). For human PBMCs, we observed multinuclear giant cells stained red in the presence of RANKL and M-CSF, and these cells were decreased by treatment with high concentrations of GlcN and COS but not by low concentrations of them (Figure 3a). In the absence of RANKL and M-CSF, these cells were not observed. For quantification, TRAP-positive multinuclear cells that stained red were counted (Figure 3b). We found that high concentrations of GlcN and COS significantly reduced the number of formed osteoclasts, whereas low concentrations of these saccharides had no significant effect. We observed similar effects of these aminosaccharides on the formation of TRAP-positive multinuclear cells in murine RAW264 cells (Figure 3c,d).

### 3.4. Effects of GlcN and COS on Actin Ring Formation of Human PBMCs

As osteoclasts form a characteristic structure called the actin ring, we evaluated the effect of saccharides on M-CSF- and RANKL-dependent actin ring formation in human PBMCs (Figure 4). We stained the actin (green) and nuclei (blue) of the differentiated cells and observed multiple giant cells with the ring-like structure of actin in the presence of RANKL and M-CSF (Figure 4b). The number of cells with such structures was decreased by treatment with high concentrations of GlcN and COS and was not observed in the absence of RANKL and M-CSF. To quantify this, the area surrounded by an actin ring with three or more nuclei was measured in each image (Figure 4c). Only high concentrations of GlcN and COS decreased the area.

## 4. Discussion

In the present study, we investigated the effects of high (500 μg/mL) and low (0.5 μg/mL) concentrations of GlcN and COS on osteoclastic differentiation in human PBMCs and murine RAW264 cells because the effects of these sugars on differentiation may vary depending on the concentration. Particularly in the case of COS, low concentrations promote differentiation, and high concentrations inhibit differentiation in in vitro mouse primary cultures [14,15]. Additionally, considering the potential variations in the effects of different cell types on differentiation, we assessed the effects of the aminosaccharides on differentiation using two model systems: human PBMCs and mouse RAW264 cells. GlcN and COS had little effect on cell viability (Figure 1). High concentrations, but not low concentrations, of GlcN and COS suppressed RANKL- (and M-CSF)-dependent increases in TRAP enzyme activity (Figure 2) and the formation of TRAP-positive multinuclear osteoclasts (Figure 3) in human PBMCs and murine RAW264 cells. In human PBMCs, high concentrations of these saccharides suppressed the formation of an actin ring (Figure 4). These results suggested that chitosan degradation products, at least, do not possess osteoclast-inducing properties. To the best of our knowledge, this is the first study to compare the effects of GlcN and COS on osteoclast differentiation at significantly different concentrations and to investigate the effect of COS on the differentiation of human PBMCs.

Our results are consistent with the report by Li et al. [14] that showed a high concentration of COS suppressed osteoclast differentiation and the report that showed the injection of low-molecular-weight COS suppressed skull resorption induced by lipopolysaccharides in vivo [22]. Lamberti et al. reported that treatment with 100 or 200 µg/mL of GlcN did not affect actin ring formation in PBMCs from healthy individuals, while it suppressed actin ring formation in PBMCs from patients with osteoarthritis [23]. In that study, they conducted experiments at concentrations lower than those yielding effects in the present study (500 µg/mL). Therefore, it cannot be ruled out that higher concentrations might have effects even in PBMCs from healthy individuals. Nevertheless, at the very least, our present findings align with theirs in confirming that GlcN treatment does not promote osteoclast differentiation. However, there is a discrepancy between our results and those reported by Bai et al., where low concentrations of COS promoted osteoclast differentiation in primary mouse cultures [15]. As Bai et al. reported that pretreatment with a low concentration of COS promotes RANKL-dependent phosphorylation of MAP kinases, such as p38, we also assessed the effect of pretreatment with a low concentration of COS on RANKL-dependent p38 phosphorylation in RAW264 cells. However, we found that COS tended to suppress p38 phosphorylation (unpublished observations). This discrepancy between our results and those reported by Bai et al. might be due to differences in the COS or cells used. Several key parameters are used to characterize chitosan: the degree of polymerization (DP) represents the number of sugar units within the chitosan molecule, with a lower DP indicating shorter molecular chains, and the degree of deacetylation (DDA) denotes the proportion of acetyl groups removed [12,24]. Most commercial chitosan polysaccharides have molecular weights ranging from 50 to 2000 kDa with DDA of commonly 80–90% [24]. Chitosan molecules with a DP < 20 are called COS [12]. As COS is made from natural products, its composition may vary depending on the manufacturer from which it is purchased. In the present study, we used COS consisting mainly of a DP = 2–6, obtained from Tokyo Chemical Industry (TCI). Li et al. used chitosan oligosaccharides with an average molecular weight of 1550 [14], which could be considered a relatively close molecular weight to that of the COS we used. Meanwhile, because chitosan with a DP < 20 is generally called a chitosan oligosaccharide, Bai et al. may have used longer chitosan oligosaccharides with a higher DP. Furthermore, the effects of other factors such as the DDA of the COS cannot be excluded. Therefore, experiments using longer COS and experiments using organically synthesized COS with controlled DDA may be crucial for future investigations.

The precise molecular mechanisms underlying the suppressive effects of high concentrations of GlcN and COS on osteoclast differentiation remain unclear. However, it can be hypothesized that COS suppresses osteoclast differentiation through its superoxide radical scavenging activity. Reactive oxygen species is important for osteoclast differentiation, and their deprivation suppresses differentiation [25,26]. Chitosan polysaccharides exhibit antioxidant activities [27]. The superoxide radical scavenging activity of chitosan increased as its chain length decreased: the superoxide radical scavenging activities of chitosan with molecular weights of 760, 120, 60, 20, and 9 k were compared, revealing that the chitosan with a lower molecular weight (shorter chain length) exhibited stronger scavenging effects [28]. However, this rule does not apply to COS with molecular weights below ~3000. COS with a molecular weight of 1100 exhibits higher radical scavenging activity compared to that with a molecular weight of 500 [29]. Additionally, for COS with molecular weights below 3000, when the DP is less than 2, the activity is particularly low [30]. As the DP of the COS used in this study was 2–6 and its average molecular weight was estimated to be ~1000, the COS used in this study is presumed to possess a certain level of radical scavenging activity, and it is speculated that this activity may exert an inhibitory effect on osteoclast differentiation.

Meanwhile, the radical scavenging activity of the glucosamine monomer is considered to be low, and it possibly affects osteoclast differentiation by other mechanisms, for example, protein *O*-GlcNAcylation. Once taken up by the cells, glucosamine is converted to UDP-GlcNAc via the hexosamine biosynthesis pathway [31]. UDP-GlcNAc is utilized as a substrate for the biosynthesis of *N*-linked glycans, *O*-linked glycans, and glycosaminoglycans, and serves as a substrate for the *O*-GlcNAc modification of proteins [31]. Nakajima et al. clearly showed that, via a glycomics approach using a mass isotopomer of glucosamine (^13^C_2_-glycosamine), extracellularly added glucosamine was taken up and metabolized within the cell and incorporated into *N*-linked and *O*-linked glycans in the mouse hepatoma cell line Hepa1-6 and the mouse pancreatic insulinoma cell line Min6 [32]. In these cell lines, externally added glucosamine was not significantly incorporated into *O*-GlcNAc-modified proteins [32]. However, in mouse RAW264 cells, the addition of GlcN or its acetylated form, GlcNAc, to the culture media has been reported to increase *O*-GlcNAc-modified proteins [16,20]. As protein *O*-GlcNAc modification is dynamically regulated in the process of osteoclast differentiation, and the promotion or suppression of *O*-GlcNAcylation suppresses differentiation [20,33,34,35], the suppressive effect of high concentrations of GlcN on differentiation in this study could be in part due to the promotion of *O*-GlcNAc modification. However, considering that the added glucosamine is utilized in UDP-GlcNAc synthesis within the cells [31], it is likely that, in human PBMCs and murine RAW264 cells as well, the synthesis of sugar chains other than *O*-GlcNAc is affected. The addition of glucosamine has been reported to alter glycosylation patterns in RAW264 cells [16]. This suggests that glucosamine may also influence differentiation by altering sugar chain synthesis pathways.

GlcN and COS are approved as pharmaceuticals and health supplements in various countries around the world. Notably, the concentrations of GlcN and COS used which affect the differentiation in this study are significantly higher than the concentrations of orally administrated GlcN in humans: when orally administered to humans at a dose of 1500 mg/day, GlcN has been reported to achieve a blood concentration of approximately 10 μM, equivalent to 2 μg/mL [36]. However, some effects can occur when GlcN is orally administered in vivo: for instance, by orally administering 1500 mg of glucosamine sulfate per day, substantial and convergent evidence has been obtained indicating a significant alleviation of symptoms related to degenerative joint disease [37]; the oral administration of GlcNAc, an *N*-acetylated form of GlcN, to mice resulted in alterations in cellular glycan structures and demonstrated therapeutic benefits in treating experimental autoimmune encephalomyelitis [38]; additionally, the oral administration of GlcNAc to human subjects showed inhibitory effects on inflammatory markers and neurodegeneration in multiple sclerosis [39]. In this study, little impact on differentiation was observed when sugars were added at concentrations (0.5 μg/mL) lower than the blood concentration of orally administered GlcN. Therefore, the possibility that GlcN and COS influence osteoclast differentiation upon oral administration to humans cannot be completely ruled out. Additionally, studies have reported that the long-term oral administration of GlcN in mice suppresses osteoclast formation at a concentration 10-fold higher than the recommended daily allowance for humans [40]. In the future, it will be essential to fine-tune the concentrations of these sugars and confirm their impact on differentiation in vitro. Additionally, because cells are exposed to these aminosaccharides before the induction of differentiation when administered orally, analyzing their effects on differentiation after pretreatment with these aminosaccharides in in vitro experiments is necessary to gain further insights into their influence.

## 5. Conclusions

In this study, we investigated whether chitosan degradation products promoted osteoclast differentiation. Specifically, we examined the impact of chitosan degradation products, COS and GlcN, on osteoclast differentiation in two models: human PBMCs and mouse macrophage-like RAW264 cells. Chitosan degradation products did not promote osteoclast differentiation, and high concentrations of these products suppressed differentiation. These findings provide valuable clinically significant information for the potential use of chitosan and its derivatives as bone-forming materials.

## Figures and Tables

**Figure 1 biotech-13-00006-f001:**
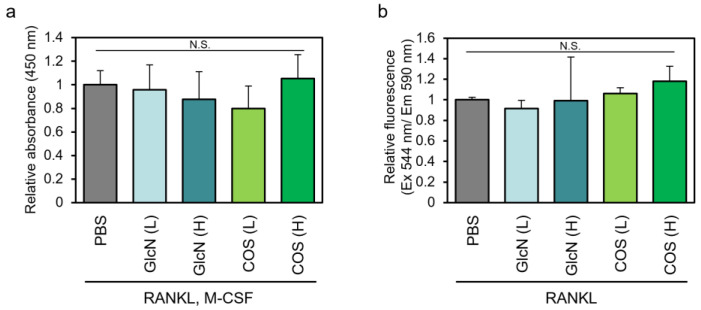
Effects of GlcN and COS on cell viability of human PBMCs and murine RAW264 cells. (**a**) Human normal PBMCs were treated with M-CSF and RANKL in the presence of low (0.5 μg/mL; (L)) or high (500 μg/mL; (H)) concentration of GlcN or COS for 10 days. The cells were then subjected to cell viability assay using Cell Counting Kit-8 reagent. (**b**) Murine RAW264 cells were treated with RANKL in the presence of low (0.5 μg/mL; (L)) or high (500 μg/mL; (H)) concentration of GlcN or COS for 4 days. The cells were then subjected to cell viability assay using PrestoBlue^TM^ reagent. Data are expressed as mean ± S.D (n = 8 or 4). N.S. means not significant.

**Figure 2 biotech-13-00006-f002:**
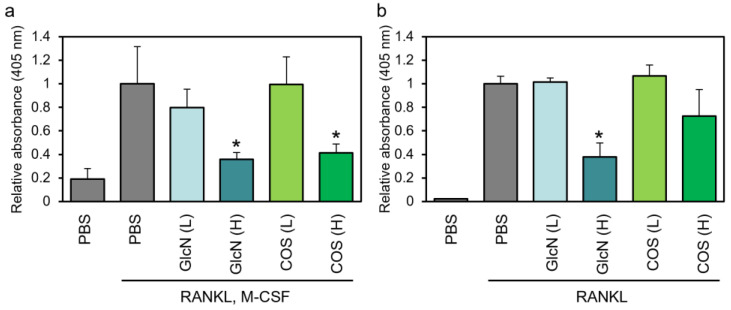
Effects of GlcN and COS on TRAP enzyme activity of human PBMCs and murine RAW264 cells. (**a**) Human normal PBMCs and (**b**) murine RAW264 cells were treated with RANKL (and M-CSF) in the presence of low (0.5 μg/mL; (L)) or high (500 μg/mL; (H)) concentration of GlcN or COS for 10 days (PBMCs) or 4 days (RAW264 cells). The cells were then subjected to TRAP enzyme activity assay. Data are expressed as mean ± S.D (n = 8 or 4). * *p* < 0.05 vs. control (PBS with RANKL (and M-CSF)).

**Figure 3 biotech-13-00006-f003:**
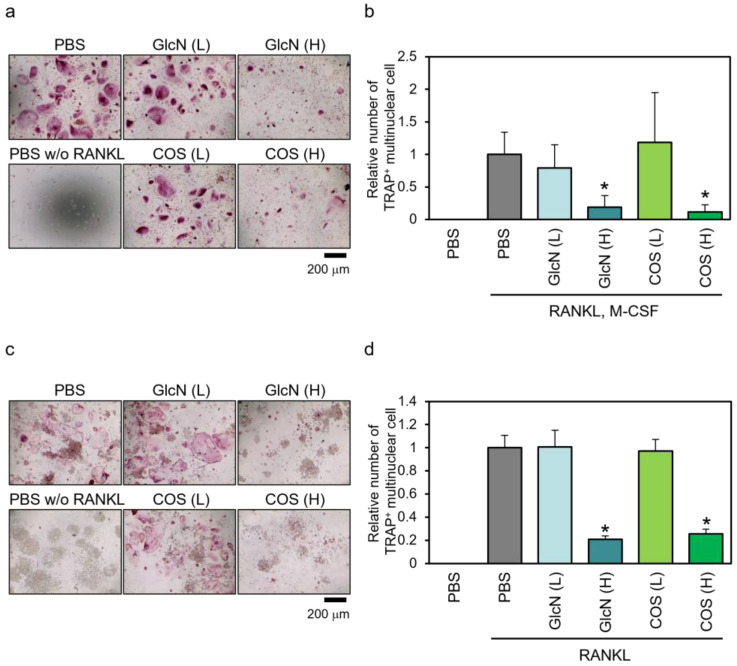
Effects of GlcN and COS on TRAP-positive multinuclear cell formation of human PBMCs and murine RAW264 cells. (**a**,**b**) Human normal PBMCs and (**c,d**) murine RAW264 cells were treated with RANKL (and M-CSF) in the presence of low (0.5 μg/mL; (L)) or high (500 μg/mL; (H)) concentration of GlcN or COS for 10 days (PBMCs) or 4 days (RAW264 cells). The cells were then subjected to TRAP staining. TRAP-positive multinuclear cells stained red were observed under a microscope with ×10 objective (**a**,**c**), and the number of formed TRAP-positive multinuclear cells was counted (**b**,**d**). Data are expressed as mean ± S.D (n = 8 or 4). * *p* < 0.05 vs. control (PBS with RANKL (and M-CSF)).

**Figure 4 biotech-13-00006-f004:**
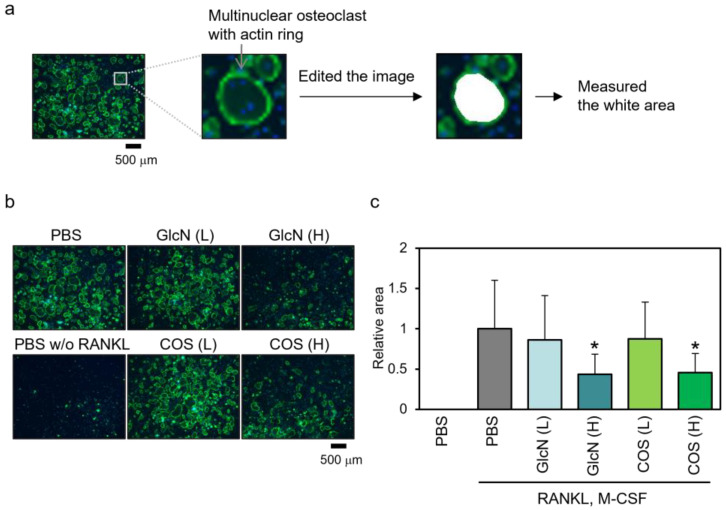
Effects of GlcN and COS on actin ring formation of human PBMCs. (**a**) Osteoclasts forming actin rings and a schematic representation for quantification. (**b**,**c**) Human normal PBMCs were treated with RANKL and M-CSF in the presence of low (0.5 μg/mL; (L)) or high (500 μg/mL; (H)) concentration of GlcN or COS for 10 days. The cells were then subjected to actin staining. Actin was stained with ActinGreen ready probe reagent (Green) and nuclei were stained with DAPI (Blue). Fluorescently stained cells were observed under a fluorescent microscope with ×4 objective (**b**), and the area surrounded by an actin ring with three or more nuclei of each image was measured using Image J software (**c**). Data are expressed as mean ± S.D (n = 8). * *p* < 0.05 vs. control (PBS with RANKL and M-CSF).

## Data Availability

The data generated or analyzed during this study are included in this article.

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
