# Peer review of "Effect of Chitosan Degradation Products, Glucosamine and Chitosan Oligosaccharide, on Osteoclastic Differentiation"

_biotech, 2024, doi:10.3390/biotech13010006_

Round 1

Reviewer 1 Report

Comments and Suggestions for Authors

For the manuscript "Effect of chitosan degradation products, glucosamine and chitosan oligosaccharide, on osteoclastic differentiation", this study investigated whether chitosan degradation products promoted osteoclast differentiation. The author examined the impact of chitosan degradation products, COS and GlcN, on osteoclast differentiation in two models: human PBMCs and mouse macrophage-like RAW264 cells. The results showed that chitosan degradation products did not promote osteoclast differentiation, and high concentrations of these products suppressed differentiation. These findings provide valuable information for the potential use of chitosan and its derivatives as bone-forming materials. However, there have some questions in your manuscript. The comments were as followed:

1.     In introduction and methods part, relevant citations are lack in both parts, please add.

2.     The format of the manuscript need to be improved, especially the references.

3.     Why these two cell models were chosen for experiments? And the relevant research on these two models also needs to be explained in the introduction section.

4.     The purpose and significance of the study also need to be written in the last paragraph of the introduction section.

5.     All figures need to be analyzed for significance, please check and modify.

6.     Why choose this high and low concentration, and whether the concentration is μg/mL or mg/mL? Please check the whole manuscript.

Author Response

RESPONSES TO COMMENTS BY REVIEWER #1

We thank you for your favorable and thoughtful comments on our manuscript, which have enriched the manuscript and produced a better and more balanced account of the research. Our responses are described below, and the modifications have been highlighted in yellow in the revised manuscript. We hope that the revised manuscript is now suitable for publication in BioTech.

Comment #1:

In introduction and methods part, relevant citations are lack in both parts, please add.

Response:

According to your suggestion, we have added relevant citations in introduction and methods part in Lines 33, 49, 59, 67, 145, and 156.

Comment #2:

The format of the manuscript need to be improved, especially the references.

Response:

According to your suggestion, we have checked and updated the format of the references.

Comment #3:

Why these two cell models were chosen for experiments? And the relevant research on these two models also needs to be explained in the introduction section.

Response:

Thank you for the suggestion. We used two cell models because the effect of GlcN and COS might be different between the cells used. We chose human PBMC as a precursor of human osteoclast that well reflects differentiation under physiological conditions, and the RAW264 cell line as a commonly used model cell line for osteoclast differentiation. Relevant research manuscripts of RAW264 cell have been cited as Reference 14, 15 in Line 93. We have modified the description and added relevant explanation (Lines 95-98).

Comment #4:

The purpose and significance of the study also need to be written in the last paragraph of the introduction section.

Response:

According to your suggestion and considering the flow of the explanation, we have added the description about the purpose and significance of the present study in the last paragraph (from line 87) and at the end of the last second paragraph (Lines 82-86) of the introduction section.

Comment #5:

All figures need to be analyzed for significance, please check and modify.

Response:

Thank you for your comment. We apologize for any confusion caused by our insufficient description. Statistical analysis had been conducted for all figures. Asterisks indicating significance had already been added to Figures 2-4. However, as there was no statistical significance among the data in Figure 1, asterisks had not been included. For Figure 1, instead of using asterisks, we have added 'N.S.' (Not Significant) and revised the figure legend.

Comment #6:

Why choose this high and low concentration, and whether the concentration is μg/mL or mg/mL? Please check the whole manuscript.

Response:

Thank you for the suggestion. The concentrations are μg (micro gram)/mL not mg (milli gram)/mL. It appeared that where we described ‘μ (micro)’ using Symbol font in the word file of our submitted manuscript were mistakenly converted to ‘m’ in the PDF file. Meanwhile, those using the Palatino Linotype font remained correctly displayed as ‘μ (micro).’ We apologize for any confusion caused by this oversight in the conversion process. We have corrected these typographical errors in the revised manuscript.

                  As to the reason why we chose the concentrations, since the treatment with low concentration of COS (0.5 μg (500 ng)/mL) was reported to promote osteoclast differentiation as described in Lines 76-78, we chose 0.5 μg/mL as ‘Low concentration.’ On the other hand, high concentration (440 μg/mL) of COS was reported to suppress osteoclast differentiation as described in Lines 73-76. As a neat and rounded concentration, we chose 500 μg/mL as ‘High concentration.’ We have added relevant description in Lines 89-94 and 121-122.

Reviewer 2 Report

Comments and Suggestions for Authors

Comments to the Authors of manuscript number: biotech-2809082 entitled “Effect of chitosan degradation products, glucosamine and chitosan oligosaccharide, on osteoclastic differentiation”.

Chitosan, a natural polysaccharide from crustaceans, is known for its biocompatibility and antibacterial properties. This study investigated the effects of chitosan degradation products, glucosamine (GlcN) and chitosan oligosaccharide (COS), on osteoclasts. High concentrations suppressed osteoclastic differentiation, indicating the safe potential use of chitosan in bone tissue repair.

1. L 60- what enzymes? Name it please

2. L 61-62- the idea is not finished?

3. L 62- and what? Why it is important? The explanation is needed

4. introduction: the text includes repetitive information, such as the repeated mention of chitosan's properties and applications. This redundancy could be eliminated for a more concise presentation.; It jumps between topics without a smooth transition, making it challenging for readers to follow the logical flow of information.;

5. The manuscript delves into technical details about chitosan, its molecular structure, and various applications without providing a clear link to the main focus of the study, which is the effect of chitosan degradation products on osteoclasts. The introduction should more directly set the stage for the study.

6. unnecessary details should be omitted to maintain focus on the main research question.

7. Some sentences are incomplete, and there are instances where information is abruptly introduced without proper context or explanation.

8. L 90-95 – the lack of hypothesis and real goal of the study. The last sentences should be removed. It is the introduction

9. If murine cell are used, it should be mentioned L 53- that bone are important not only for humans

10. L 90-93 – to materials should be moved

11. 2.2: The text mentions the treatment with 0.5 or 500 mg/mL of D(+)-glucosamine hydrochloride (GlcN) or chitosan oligosaccharide (COS) for both cell types but does not elaborate on the significance of these concentrations. A brief explanation of the reasoning behind choosing these concentrations or their relevance to the study objectives would be useful.; The text mentions the differentiation period (10 days for PBMCs and 4 days for RAW264 cells), but it would be helpful to provide a rationale for the chosen duration. Why were these specific timeframes selected, and what biological processes are expected to occur during this time?

12. While the concentrations of GlcN and COS are provided, the units are inconsistent (mg/mL for GlcN and μg/mL for COS). Standardizing the units for clarity would be advisable.

13. It would be useful to mention whether there was a control group without the addition of GlcN or COS to serve as a baseline for comparison.

14. 2.3: The SpectraMax M5 microplate reader is mentioned for measuring fluorescence or absorbance. While this provides information on the equipment used, specifying any instrument settings or parameters (e.g., gain, integration time) might be relevant for reproducibility. The text does not specify the units of measurement for fluorescence or absorbance. Including this information would ensure clarity in reporting the assay results.; It would be beneficial to mention whether any quality control measures were implemented during the assay, such as positive and negative controls.; Providing information on the number of replicates or independent experiments conducted for the cell viability assay would contribute to the assessment of the assay's reproducibility.

15. 2.4: The text does not specify the units of measurement for absorbance.

16. 2.5: TRAP-positive cells are those that stain red and contain three or more nuclei. Clarifying the significance of these criteria in identifying osteoclasts and ensuring consistency in cell counting would be beneficial.

17. 2.6: Images are captured and analyzed using Image J software. Elaborating on the specific parameters or measurements obtained from the analysis, and the significance of selecting the area surrounded by actin rings with three or more nuclei, would add depth to the description.

18. It's good that Tukey's multiple comparison test is mentioned. Including a short sentence explaining why this test was chosen, especially in the context of your experimental design, would be beneficial.; If applicable, consider providing exact P-values rather than stating P < 0.05. This might be relevant if there are borderline significance levels or if readers need precise information about the statistical significance.

19. The discussion provides a comprehensive overview of the experimental findings.

The comparison of effects between high and low concentrations of GlcN and COS is well-explained, offering a nuanced understanding.

The discussion of variations in results from other studies, such as Bai et al., is insightful and adds depth to the interpretation.

20. The emphasis on this study being the first to compare the effects of GlcN and COS at significantly different concentrations, as well as the investigation of COS on human PBMCs, highlights the novelty of the research.

21. The discussion delves into potential molecular mechanisms, particularly the radical-scavenging activity of COS and the influence of glucosamine on protein O-GlcNAcylation and glycosylation patterns. This adds depth to the interpretation of results.

22. The consideration of concentrations used in the study compared to those achieved through oral administration in humans is a valuable addition. The discussion appropriately highlights the need for fine-tuning concentrations for in vitro experiments to mimic in vivo conditions more accurately.

23. The discussion appropriately identifies future research directions, including the need for experiments using longer COS and organically synthesized COS with controlled DDA. This demonstrates a forward-looking perspective.

24. The discussion is well-organized and follows a logical flow, connecting experimental findings, discrepancies with other studies, potential mechanisms, and clinical implications.; The discussion effectively summarizes the key findings and their implications, providing a concise conclusion to the study.

25. The discussion is thorough, insightful, and well-structured. It successfully integrates experimental findings with relevant literature, addresses discrepancies, and proposes plausible mechanistic explanations. The consideration of clinical relevance and future directions enhances the overall quality of the discussion.

Author Response

RESPONSES TO COMMENTS BY REVIEWER #2

We thank you for your favorable and thoughtful comments on our manuscript, which have enriched the manuscript and produced a better and more balanced account of the research. Our responses are described below, and the modifications have been highlighted in yellow in the revised manuscript. We hope that the revised manuscript is now suitable for publication in BioTech.

Comment #1:

L 60- what enzymes? Name it please

Response:

We apologize for the unclear statement. Chitosan has been reported to be degraded primarily by lysozyme. The possibility that other enzymes may also be involved cannot be ruled out, but it has not been clear. Therefore, we have corrected the description to "mainly by lysozyme"(Line 69).

Comment #2:

L 61-62- the idea is not finished?

Response:

Thank you for the suggestion. As you pointed out, the idea (Lines 70-72 in the revised manuscript) has not been finalized. Considering the overall flow of the explanation, we have revised the content of the Introduction section overall and relocated the description of osteoclasts, initially present from Line 64 in our previously submitted manuscript, to precede the existing content.

Comment #3:

L 62- and what? Why it is important? The explanation is needed

Response:

Thank you for the suggestion. We have emphasized the importance at the end of the paragraph (Lines 82-86).

Comment #4:

introduction: the text includes repetitive information, such as the repeated mention of chitosan’s properties and applications. This redundancy could be eliminated for a more concise presentation.; It jumps between topics without a smooth transition, making it challenging for readers to follow the logical flow of information.

Response:

Thank you for your comment. Taking into consideration this and the feedback provided in your comment #2, we have revised the flow of the explanation and omitted repetition. Specifically, as described in the response to comment #2, we have summarized the relationship between chitosan, its degradation products, and osteoclasts (Lines 68-86). Additionally, we have consolidated the descriptions related to bone and osteoclasts (Lines 46-67).

Comment #5:

The manuscript delves into technical details about chitosan, its molecular structure, and various applications without providing a clear link to the main focus of the study, which is the effect of chitosan degradation products on osteoclasts. The introduction should more directly set the stage for the study.

Response:

Thank you for the suggestion. Taking your suggestion into account and considering the flow of the explanation, we have made overall revisions to the introduction. We have integrated the first and second paragraphs of originally submitted manuscript and have relocated the technical details regarding chitosan to the Discussion section (Lines 279-284). In addition, we have extensively revised and supplemented the latter part of the Introduction to ensure a clear link to the main focus of this research (Lines 82-96).

Comment #6:

unnecessary details should be omitted to maintain focus on the main research question.

Response:

As described in the response to the comment #5, we have revised the introduction.

Comment #7:

Some sentences are incomplete, and there are instances where information is abruptly introduced without proper context or explanation.

Response:

As described in the response to the comments #2, #3, #4, and #5, we have revised the introduction.

Comment #8:

L 90-95 – the lack of hypothesis and real goal of the study. The last sentences should be removed. It is the introduction.

Response:

Thank you for the suggestion. As recommended, we included descriptions of the study's significance and purpose in the last paragraph (Lines 87-88) and the end of last second paragraph (Lines 82-86) of the introduction. Additionally, we have removed the last sentence in the originally submitted manuscript's last paragraph of the introduction.

Comment #9:

If murine cell are used, it should be mentioned L 53- that bone are important not only for humans.

Response:

Thank you for the suggestion. We have replaced ‘human body’ with ‘body’ in Line 46.

Comment #10:

L 90-93 – to materials should be moved.

Response:

According to your suggestion, as described in the response to the comment #8, we have removed the last sentence in the originally submitted manuscript's last paragraph of the introduction.

Comment #11:

2.2: The text mentions the treatment with 0.5 or 500 mg/mL of D(+)-glucosamine hydrochloride (GlcN) or chitosan oligosaccharide (COS) for both cell types but does not elaborate on the significance of these concentrations. A brief explanation of the reasoning behind choosing these concentrations or their relevance to the study objectives would be useful.; The text mentions the differentiation period (10 days for PBMCs and 4 days for RAW264 cells), but it would be helpful to provide a rationale for the chosen duration. Why were these specific timeframes selected, and what biological processes are expected to occur during this time?

Response:

Thank you for the suggestions. As to the reason why we chose the concentrations, since the treatment with low concentration of COS (0.5 μg (500 ng)/mL) was reported to promote osteoclast differentiation as described in Lines 76-78, we chose 0.5 μg/mL as ‘Low concentration.’ On the other hand, high concentration (440 μg/mL) of COS was reported to suppress osteoclast differentiation as described in Lines 73-76. As a neat and rounded concentration, we chose 500 μg/mL as ‘High concentration.’ To emphasize this, we have added the relevant descriptions in Lines 89-95 and 121-122.

                     For the duration of the differentiation process, PBMCs originally exist as monocytes, which undergo differentiation into macrophages and eventually progress to osteoclasts. Consequently, there is a tendency for a longer induction time compared to RAW264 cells, which inherently possess macrophage-like characteristics. To emphasize this point, the description “during which monocytes were differentiated to macrophages and subsequently to osteoclasts” has been added in Lines 117-118.

Comment #12:

While the concentrations of GlcN and COS are provided, the units are inconsistent (mg/mL for GlcN and μg/mL for COS). Standardizing the units for clarity would be advisable.

Response:

Thank you for the suggestions. The concentrations are μg (micro gram)/mL not mg (milli gram)/mL. It appeared that where we described ‘μ (micro)’ using Symbol font in the word file of our submitted manuscript were mistakenly converted to ‘m’ in the PDF file. Meanwhile, those using the Palatino Linotype font remained correctly displayed as ‘μ (micro).’ We apologize for any confusion caused by this oversight in the conversion process. We have corrected these typographical errors in the revised manuscript.

Comment #13:

It would be useful to mention whether there was a control group without the addition of GlcN or COS to serve as a baseline for comparison.

Response:

Thank you for the suggestion. We have added relevant descriptions in Lines 120-121, 126 and 137-138.

Comment #14:

2.3: The SpectraMax M5 microplate reader is mentioned for measuring fluorescence or absorbance. While this provides information on the equipment used, specifying any instrument settings or parameters (e.g., gain, integration time) might be relevant for reproducibility. The text does not specify the units of measurement for fluorescence or absorbance. Including this information would ensure clarity in reporting the assay results.; It would be beneficial to mention whether any quality control measures were implemented during the assay, such as positive and negative controls.; Providing information on the number of replicates or independent experiments conducted for the cell viability assay would contribute to the assessment of the assay’s reproducibility.

Response:

Thank you for the suggestions. Following your advice, we incorporated details about the analysis settings of the SpectraMax M5 microplate reader in Lines 133-134. Additionally, we included information about the wavelength used for fluorescence and absorbance measurements in the bar graphs of Figures 1 and 2.

               Regarding the control, as described in the response to your comment #13, we have added the relevant descriptions in Lines 120-121, 126 and 137-138. In addition, we have added description about the reproducibility in Lines 176-177.

Comment #15:

2.4: The text does not specify the units of measurement for absorbance.

Response:

Thank you for the feedback. However, as absorbance generally lacks specific units, we cannot provide a unit designation. Regarding the wavelength, as described in the response to your comment #14, we have included this information in the figures.

Comment #16:

2.5: TRAP-positive cells are those that stain red and contain three or more nuclei. Clarifying the significance of these criteria in identifying osteoclasts and ensuring consistency in cell counting would be beneficial.

Response:

In the case of proliferating cells, nuclei may appear as two due to ongoing division. To specifically count multinucleated cells and exclude those undergoing division, it is common practice to count cells with three or more nuclei when quantifying multinucleated osteoclasts. To emphasize this point, we have added related description in Lines 155-156.

Comment #17:

2.6: Images are captured and analyzed using Image J software. Elaborating on the specific parameters or measurements obtained from the analysis, and the significance of selecting the area surrounded by actin rings with three or more nuclei, would add depth to the description.

Response:

Thank you for the suggestion. We have manually selected the area surrounded by actin rings. Therefore, we could not describe the specific parameters. We have described the method in Figure 4a and added related description in Line 170.

Comment #18:

It’s good that Tukey’s multiple comparison test is mentioned. Including a short sentence explaining why this test was chosen, especially in the context of your experimental design, would be beneficial.; If applicable, consider providing exact P-values rather than stating P < 0.05. This might be relevant if there are borderline significance levels or if readers need precise information about the statistical significance.

Response:

Thank you for the suggestion. We have added relevant description in Line 175. Regarding exact P-values, our software used does not provide precise values. Therefore, we refrain from reporting exact P-values in this study.

Comment #19:

The discussion provides a comprehensive overview of the experimental findings.

The comparison of effects between high and low concentrations of GlcN and COS is well-explained, offering a nuanced understanding.

The discussion of variations in results from other studies, such as Bai et al., is insightful and adds depth to the interpretation.

Response:

Thank you for your favorable comment.

Comment #20:

The emphasis on this study being the first to compare the effects of GlcN and COS at significantly different concentrations, as well as the investigation of COS on human PBMCs, highlights the novelty of the research.

Response:

Thank you for your favorable comment.

Comment #21:

The discussion delves into potential molecular mechanisms, particularly the radical-scavenging activity of COS and the influence of glucosamine on protein O-GlcNAcylation and glycosylation patterns. This adds depth to the interpretation of results.

Response:

Thank you for your favorable comment.

Comment #22:

The consideration of concentrations used in the study compared to those achieved through oral administration in humans is a valuable addition. The discussion appropriately highlights the need for fine-tuning concentrations for in vitro experiments to mimic in vivo conditions more accurately.

Response:

Thank you for your favorable comment.

Comment #23:

The discussion appropriately identifies future research directions, including the need for experiments using longer COS and organically synthesized COS with controlled DDA. This demonstrates a forward-looking perspective.

Response:

Thank you for your favorable comment.

Comment #24:

The discussion is well-organized and follows a logical flow, connecting experimental findings, discrepancies with other studies, potential mechanisms, and clinical implications.; The discussion effectively summarizes the key findings and their implications, providing a concise conclusion to the study.

Response:

Thank you for your favorable comment.

Comment #25:

The discussion is thorough, insightful, and well-structured. It successfully integrates experimental findings with relevant literature, addresses discrepancies, and proposes plausible mechanistic explanations. The consideration of clinical relevance and future directions enhances the overall quality of the discussion.

Response:

Thank you for your favorable comment.

Reviewer 3 Report

Comments and Suggestions for Authors

The results show that the chitosan degradation products D-glucosamine, either as monomeric form (GlcN) or as oligomer (polymerization 2 to 6 units, COS), that probably accompanying chitosan, do not possess osteoclast-inducing or osteoclast differentiation properties. This reinforces the use of chitosan its composite materials for bone tissue repair.

The effect has been tested at two extreme concentrations, low (L) and high (H) and the cell models have been human PBMCs and murine macrophage-like cell line.        

According to the authors, a parallel study at two very different concentrations has never been carried out before, although some previous related data can be found (Refs. 15,16, 22). So, the study is interesting and valuable, but it is not totally novel, and some points are still unclear. It would be interesting the effect on complete transduction pathways, as p38, but these results are not revealed by the authors (line 263). The will of the authors would be accepted concerning this point.

 Points to be addressed before acceptance:

The supposedly concentrations for GlcN and COS are high (500 μg/mL) and low (0.5 μg/mL), but this point should be checked. At line 167 (first time) and legends of all figures 1, 2, 3, 4 is stated that the concentrations tested were 0.5 and 500 mg/mL, that is 1000-fold higher than in the abstract. In turn in some other sections, the concentration is given as nanograms (i.e. line 83, 500 ng/mL), that really is 0.5 μg/mL. At the current form, I find this is a mess as the comparison is between 0.5 vs 500. In sum, concentrations and units should be checked throughout the manuscript and the use of the same units in all sections is recommended for avoiding confusion.

Line 101: The mouse macrophage-like RAW264. The “macrophage-like”  nature of the RAW264 cell line would be added to the abstract (line 17) for a better comprehensive reading of that section.

Indicate the wavenumber of the absorbance measurements at Figure 2

Recommendation: COS is sometimes referred as saccharide and oligosaccharide. It would be convenient that the term was aminosaccharide, as it is an oligomeric chain of D-glucosamine. In fact, the results obtained in most of the figures for GlcN and COS are very similar (only Figure 2b shows a statistically significant difference).

Author Response

RESPONSES TO COMMENTS BY REVIEWER #3

We thank you for your favorable and thoughtful comments on our manuscript, which have enriched the manuscript and produced a better and more balanced account of the research. Our responses are described below, and the modifications have been highlighted in yellow in the revised manuscript. We hope that the revised manuscript is now suitable for publication in BioTech.

Comment #1:

The supposedly concentrations for GlcN and COS are high (500 μg/mL) and low (0.5 μg/mL), but this point should be checked. At line 167 (first time) and legends of all figures 1, 2, 3, 4 is stated that the concentrations tested were 0.5 and 500 mg/mL, that is 1000-fold higher than in the abstract. In turn in some other sections, the concentration is given as nanograms (i.e. line 83, 500 ng/mL), that really is 0.5 μg/mL. At the current form, I find this is a mess as the comparison is between 0.5 vs 500. In sum, concentrations and units should be checked throughout the manuscript and the use of the same units in all sections is recommended for avoiding confusion.

Response:

Thank you for the suggestion. The concentrations are μg (micro gram)/mL not mg (milli gram)/mL. It appeared that where we described ‘μ (micro)’ using Symbol font in the word file of our submitted manuscript were mistakenly converted to ‘m’ in the PDF file. Meanwhile, those using the Palatino Linotype font remained correctly displayed as ‘μ (micro).’ We apologize for any confusion caused by this oversight in the conversion process. We have corrected these typographical errors in the revised manuscript. In addition, we have updated the concentration unit to 'μg/mL' in Lines 76 and 80.

Comment #2:

Line 101: The mouse macrophage-like RAW264. The “macrophage-like”  nature of the RAW264 cell line would be added to the abstract (line 17) for a better comprehensive reading of that section.

Response:

According to your comment, we have added “macrophage-like” in Line 17.

Comment #3:

Indicate the wavenumber of the absorbance measurements at Figure 2

Response:

Thank you for the feedback. We have included information about the wavenumber in the bar graphs of Figure 2.

Comment #4:

Recommendation: COS is sometimes referred as saccharide and oligosaccharide. It would be convenient that the term was aminosaccharide, as it is an oligomeric chain of D-glucosamine. In fact, the results obtained in most of the figures for GlcN and COS are very similar (only Figure 2b shows a statistically significant difference).

Response:

Thank you for the suggestion. As recommended, we have substituted the statements regarding COS in Lines 184, 219, 259, 365 and 367.

Reviewer 4 Report

Comments and Suggestions for Authors

The manuscript deals with the effects of chitosan degradation products, glucosamine and chitosan oligosaccharide, on osteoclastic differentiation.

-The Introduction is longer than necessary, so it can be reduced in length. Some passages can also be transferred to the Discussion.

-I have no major comments for the technical performance of the study.

-Statistics. Were they data normally distributed? This is not clarified.

-Please colorize the graphs to make them more attractive for reading and interpreting.

-The Discussion can be divided into two sections to allow easier flow of reading.

-One or two recently published relevant papers are missing. I suggest to cite them and to use their findings for discussion against those of the present study.

-In Discussion, please include a paragraph with the clinical applications of the findings.

Overall. Improvement as indicated and resubmission.

Author Response

RESPONSES TO COMMENTS BY REVIEWER #4

We thank you for your favorable and thoughtful comments on our manuscript, which have enriched the manuscript and produced a better and more balanced account of the research. Our responses are described below, and the modifications have been highlighted in yellow in the revised manuscript. We hope that the revised manuscript is now suitable for publication in BioTech.

Comment #1:

The Introduction is longer than necessary, so it can be reduced in length. Some passages can also be transferred to the Discussion.

Response:

Thank you for the suggestion. Following your feedback, we have reviewed and revised the entire Introduction section. Specifically, we have moved the information regarding the composition of chitosan, originally present in the first paragraph of the initial manuscript, to the Discussion section (Lines 288-293). Furthermore, we have integrated the first and second paragraphs of the Introduction of our previously submitted original manuscript.

Comment #2:

Statistics. Were they data normally distributed? This is not clarified.

Response:

Thank you for the suggestion. Since the sample size is not large enough, we have not confirmed whether the data follows a normal distribution; hence, statistical analyses are conducted under the assumption of normality. We have added relevant description in Line 173.

Comment #3:

Indicate the wavenumber of the absorbance measurements at Figure 2

Response:

Thank you for the feedback. We have included information about the wavenumber in the bar graphs of Figure 2.

Comment #4:

Please colorize the graphs to make them more attractive for reading and interpreting.

Response:

Thank you for the suggestion. We have colorized the graphs and replaced all figures.

Comment #5:

One or two recently published relevant papers are missing. I suggest to cite them and to use their findings for discussion against those of the present study

Response:

Thank you for the suggestion. We have incorporated references of two new studies (Lambertini et al., 2021; Asai et al., 2016) and provided additional relevant descriptions in Lines 273-280 and 360-363.

Comment #6:

In Discussion, please include a paragraph with the clinical applications of the findings.

Response:

Thank you for the suggestion. We consider that results obtained in this study is important for the clinical application for chitosan derivatives in terms of safety. However, in the present study, we did not assess the direct clinical application of GlcN or COS. Since discussing the direct clinical application of GlcN and COS using a paragraph may lead over discussion, we would like to refrain from delving extensively into this aspect. Instead, we have modified the text in the Conclusion section (Line 375) to emphasize the significance of the information that chitosan degradation products do not promote osteoclast differentiation, underscoring its relevance in clinical applications.

Round 2

Reviewer 4 Report

Comments and Suggestions for Authors

The authors have improved the manuscript in accordance with the recommendations. I have no further comments.